# Alcohol Harms over a Period of Alcohol Policy Reform: Surveys of New Zealand College Residents in 2004 and 2014

**DOI:** 10.3390/ijerph17030836

**Published:** 2020-01-29

**Authors:** Kypros Kypri, Brett Maclennan, Jennie Connor

**Affiliations:** 1Department of Preventive and Social Medicine, University of Otago, Dunedia 9016, New Zealand; brett.maclennan@otago.ac.nz (B.M.); jennie.connor@otago.ac.nz (J.C.); 2School of Medicine and Public Health, University of Newcastle, Callaghan NSW 2308, Australia

**Keywords:** alcohol, risky, binge, college, students, policy

## Abstract

**Background:** We estimated the change in the prevalence of harms attributed by students to their drinking and to others’ drinking, over a decade of concerted effort by university authorities to reduce antisocial behaviour and improve student safety. Interventions included a security and liaison service, a stricter code of conduct, challenges to liquor license applications near campus, and a ban on alcohol advertising. **Methods:** We used a pre-post design adjusting for population changes. We invited all students residing in colleges of a New Zealand University to complete web surveys in 2004 and 2014, using identical methods. We estimated change in the 4-week prevalence of 15 problems and harms among drinkers, and nine harms from others’ drinking among all respondents. We adjusted for differences in sample sociodemographic characteristics between surveys. **Results:** Among drinkers there were reductions in several harms, the largest being in acts of vandalism (7.1% to 2.7%), theft (11% to 4.5%), and physical aggression (10% to 5.3%). Among all respondents (including non-drinkers), there were reductions in unwanted sexual advances (14% to 8.9%) and being the victim of sexual assault (1.0% to 0.4%). **Conclusion:** Alcohol-related harm, including the most serious outcomes, decreased substantially among college residents in this period of alcohol policy reform. In conjunction with evidence of reduced drinking to intoxication in this population, the findings suggest that strategies to reduce the availability and promotion of alcohol on and near campus can substantially reduce the incidence of health and social harms.

## 1. Introduction

University students are renowned for ‘heavy episodic’, ‘binge’, or ‘risky single episode’ drinking, i.e., consuming enough alcohol, rapidly enough, to become intoxicated. This is often characterised as at least four or five drinks (40–60 g ethanol) within two hours. In a man of average body weight, without significant tolerance, this level of consumption produces a blood alcohol concentration between 0.05 and 0.10 g/dL [1], above the legal limit for driving in most countries. 

In studies making the comparison, which were mainly performed in the USA in the early 2000s, college students in their late teens and early 20s were found to consume a greater volume of alcohol and had a higher prevalence of consuming ≥5 drinks on a single occasion than thein non-student peers [2]. College students were also more likely to report symptoms of acute problems from their drinking (“alcohol abuse”) [3]. A New Zealand study estimated that, compared with their non-student peers, university students had twice the prevalence of scoring ≥8 (‘hazardous drinking’) and three times the prevalence of scoring ≥15 (‘harmful drinking’), on the Alcohol Use Disorders Identification Test [4].

In a previous paper we estimated changes in drinking at a New Zealand university with a long history of risky drinking and related social disorder, finding no change in the prevalence of drinking per se, but large reductions in risky consumption over a 10 year period [5]. For example, the 7-day prevalence of drinking to intoxication decreased from 45% in 2004 to 33% in 2014 (absolute difference: 12%; 95% CI: 7%, 17%) among students living in residential colleges. We cautiously attributed these changes to a combination of deliberate efforts by the university to reduce the promotion of alcohol and student misconduct [6], and market conditions that caused the closure of student pubs in the area surrounding campus [5]. These efforts and conditions were independent of more recent alcohol policy reforms in New Zealand that are unlikely to have significantly affected student drinking in 2013–2014 due to the phased implementation of the reforms (several came into effect after the 2014 survey) and the absence of a local alcohol policy in Dunedin until 2019 [7,8].

The New Zealand experience contrasts with findings from surveys of nationally representative samples of college students between 1993 and 2001 in the USA. Wechsler and colleagues [9] found no change in the prevalence of heavy episodic drinking despite participants reporting increased exposure to university interventions over this 8-year period. These interventions were primarily education-based programs on the health effects of alcohol, provision of information on seeking help for alcohol problems, dissemination of institutional alcohol rules and the penalties for breaching these, and a slight increase in the use of punitive measures (i.e., fines, community service, compulsory education program attendance and treatment referrals). Previous interventions found to be effective among university students have involved authorities external to the university increasing their enforcement of state liquor laws; however, such efforts are not applied widely or consistently over time [10,11,12].

While drinking behaviour is an important outcome by which to judge the effects of changes in policy, it is arguably a means to the end of reducing the incidence and severity of various harms. Vandalism, physical aggression, and sexual assault are what motivated efforts to make the campus and surrounding environment safer for students and other members of the community. Accordingly, in this study we estimated the change in the prevalence of a range of health, social, and legal consequences of risky drinking, in the same population, between 2004 and 2014.

## 2. Materials and Methods

### 2.1. Ethical Approval

The University of Otago Human Ethics Committee (01/117, 12/278, 14/085) approved the study protocols.

### 2.2. Surveys

We invited University of Otago students living in the 12 Dunedin residential colleges in April-May 2004 to participate in a survey using a web-survey method described previously [13]. We asked respondents about their drinking and experience of harm (see http://ipru3.otago.ac.nz/demo/halls2004/). We employed the same methods and many of the same questions in a survey of the same colleges, as well as two that opened after 2004, in April-May 2014 (see http://ipru3.otago.ac.nz/demo/halls2014/).

We sent personalised letters inviting all residents (2004: N = 2497; 2014: N = 3262) to participate by clicking on the hyperlink sent to their university e-mail account. In the following weeks, we sent up to one reminder letter and three reminder e-mails to non-respondents.

We asked respondents who had consumed alcohol in the previous four weeks (who we defined as drinkers) if they had experienced a range of problems as a result of their drinking during that time [14]. The following questions asked all respondents if they had experienced consequences in the past four weeks from other students’ drinking while in their college, e.g., “being pushed, hit or otherwise assaulted” [15].

### 2.3. Analysis

We compared proportions of respondents experiencing the outcomes of interest in each survey, using Stata/SE 13.1 [16]. To account for differences between surveys in the demographic mix of respondents and their pre-university drinking behavior we weighted estimates from the 2014 survey for changes in distributions of participant gender, age, and ethnicity, and for past-year hazardous drinking (measured with the AUDIT [17]) using logistic regression models. Employing the Stata *cluster* command, we inflated standard errors to reflect clustering of respondents within colleges, and adjusted prevalence estimates using the *margins* command [18]. 

## 3. Results

### 3.1. Response Fractions

Of the 2497 students living in the Dunedin residential colleges in 2004, 1594 (64%) completed the questionnaire and a further 68 (3%) submitted responses that met a minimum data requirement. The corresponding figures for 2014 (N = 3262) were 1835 (56%) and 106 (3%).

### 3.2. Demographic Characteristics

Table 1 shows that approximately two thirds of respondents were women and the majority were 18 years old. The proportions of men and those aged ≥19 years who responded to the 2014 survey (34% and 23%, respectively) were smaller than in 2004 (38% and 33%, respectively). The proportions of Māori (indigenous), Pacific, and first-year students responding in 2014 were greater than in 2004, while the proportions of Asian students and students beyond their first year of study were smaller.

### 3.3. Alcohol-Related Problems that Respondents Attributed to Their Own Drinking

Table 2 shows proportions of residents who had experienced problems from their own drinking in the previous four weeks. The prevalence of most problems declined, particularly the more serious categories: acts of vandalism, stealing private or public property, and being physically aggressive toward others.

### 3.4. Alcohol-Related Problems that Respondents Attributed to Others’ Drinking

Table 3 shows the proportion of residents who experienced each of nine problems at their residential college that they attributed to students who had been drinking. The prevalence of all but two of the nine problems declined from 2004 to 2014. The exceptions were “baby-sitting” a drunk student and being insulted or humiliated. The largest absolute reductions were in having study and/or sleep interrupted and unwanted sexual advances. The relative reduction in sexual assault was 60%.

### 3.5. Sensitivity Analysis 

To investigate the effect of non-response we adopted an approach previously employed to quantify non-response bias in survey data [19]. We estimated the proportion of respondents who reported the following outcomes in 2004 among those who had participated by day 21 (i.e., the day we reached 59% response, our final response rate in 2014): committing an act of vandalism, being physically aggressive towards someone, having sex later regretted, being assaulted, and experiencing an unwanted sexual advance. For each of these outcomes, the difference between the proportion of all respondents in 2004 who endorsed them and the proportion of those who responded by day 21 who endorsed them, was <1%. 

We then estimated differences in the proportion of respondents with these outcomes in 2004 versus 2014, based on the assumption that the true prevalence among non-respondents was 20%, 33%, 50%, or 100% higher than among respondents. The result was larger differences in estimates from 2004 to 2014 across the outcomes, ranging from <0.01% (pushed, hit or otherwise assaulted) to 1.3% (committing an act of vandalism; being physically aggressive towards someone). 

## 4. Discussion

The proportion of residents reporting alcohol-related problems that they attributed to their own or others’ drinking declined markedly from 2004 to 2014, while “baby-sitting” drunk students increased. The results are consistent with the anecdotal accounts of college administrators that alcohol-related disorder declined across the decade, and that residents had been encouraged to assume greater responsibility for their peers, particularly when out drinking. 

In 2004 and 2014 the Dunedin colleges housed a similar proportion of students who had consumed alcohol in the preceding calendar year (i.e., before they became residents); however, in 2014, a smaller proportion had consumed alcohol hazardously in the preceding year. This may reflect the greater proportion of 16–17 year-olds in the colleges in 2014 [5]. While New Zealand law does not prohibit alcohol consumption per se at any age, students under 18 years of age cannot legally purchase alcohol, and there is evidence that this law is effective in reducing alcohol-related harm [20]. Accordingly, this change in demographic composition of the student population may account for some of the difference observed. 

It is noteworthy that two thirds of the respondents in the 2014 survey were women. This reflects the fact that there are more women at the university and residing in the halls, and that women were more likely to respond. The study aimed to recruit a sample to represent the student population. Our previous analysis of selective non-response in this population suggests that it is likely to have biased estimates of the prevalence of alcohol-related harm downward to a small extent [21]. In relation to this, drinking patterns in this population are similar by gender [22].

Strengths of the study include the equivalent timing of the surveys during the academic year, and the use of identical questions and survey methods, facilitating comparisons over time. Sensitivity analyses suggest that the difference in response fractions is unlikely to have substantially biased the estimates of change. 

The lack of a control setting limits the strength of inference one can make about the impact of changes at the university on alcohol consumption and harms in the colleges. However, the reductions in this study are consistent with those in the wider University of Otago population, and contrast with what appears to have occurred at other New Zealand universities [5].

No significant change in alcohol policy occurred at the local government level in Dunedin during the study period. Localised strategies to improve public safety and reduce alcohol-related crime were focused on the Central Business District rather than North Dunedin, where most of the student population resides [23]. 

The blood alcohol limit for drivers under 20 years of age was reduced from 0.03 g/dL to zero in 2011. However, the residential colleges in Dunedin are within walking distance of campus and alcohol outlets [24], so residents do not rely heavily on driving. Accordingly, if there were an increase in the deterrence effect of the law on drinking per se (as opposed to drink-driving) it is unlikely to have affected enough of the student population to account for the changes observed.

Our previous research showed that drinking in the wider student population decreased at Otago compared with other universities from 2005 to 2013 [5]. We also found that student drinking and harms among Otago students decreased after the introduction of *Campus Watch* and other policy changes [6]. We argued that it was unlikely such changes merely reflected a national trend or local government interventions, but without a randomised design they cannot be safely attributed to the university’s efforts. 

The marked decrease in drinking at on-licenced premises makes the pub closures in North Dunedin a likely explanation for at least part of the observed changes, given that drinking episodes in pubs are more likely to result in intoxication, particularly among men, than is drinking in other locations [25]. Quantifying the harm associated with the closures would be problematic given that the change in exposure would be relatively small. However, the pubs that closed were “student pubs” that had been frequented by many generations of students. Their closure may have had disproportionate effects that could not be captured by a simple test of correlation.

Overall, there has been a shift among Otago students to drinking in fewer types of locations, with a marked shift away from drinking in pubs [5]. This coincided with a large reduction in the prevalence of intoxication among Otago students [5] and in harms among college residents, shown in the present study. This is consistent with what we know from previous research among New Zealand university students. In a study at five universities, Connor et al. [26] found that the risk of students experiencing harm when drinking increased as the number of types of drinking location increased, even after accounting for alcohol consumption. Other studies have suggested that drinking in pubs, residential colleges and private residences facilitates heavy drinking and intoxication among New Zealand university students relative to other locations [25,27]. 

Reducing the tendency for pub drinking may have changed a prevalent pattern of multi-location drinking that could account for the reductions in vandalism, theft, physical aggression, and sexual assault we found in this study. Students may be exposed to fewer opportunities to commit, or be victims of, such acts [28], e.g., when walking between colleges and pubs. Notably, the increase in “baby-sitting” of drunk students despite decreasing drinking to intoxication over the same period, suggests success in the colleges’ efforts to encourage students to look after each other when drinking.

The findings are broadly concordant with previous research showing that policies restricting the availability and promotion of alcohol are the most effective at reducing alcohol consumption and harm among the general population [29]. There is evidence that these policies are also effective among university students, particularly when implemented alongside other strategies, although it is not clear what the ideal package of strategies might be [10].

## 5. Conclusions

In this period of alcohol policy reform, alcohol-related harm, including the most serious outcomes, decreased substantially among college residents. In conjunction with evidence of reduced drinking to intoxication in this population, the findings suggest that strategies to reduce the availability and promotion of alcohol on and around campus can substantially reduce the incidence of health and social harms. 

## Figures and Tables

**Table 1 ijerph-17-00836-t001:** Demographic characteristics of the residential college population and respondents in 2004 and 2014.

	2004		2014	
	Population (n = 2497)	Respondents (n = 1662)	Population (n = 3261)	Respondents (n = 1941)
	*%*	*%*	*%*	*%*
**Gender**				
Female	59	62	57	66
Male	41	38	43	34
**Age**				
16–17 years	3	3	9	10
18 years	62	64	69	67
19 years or older	35	33	21	23
**Ethnicity**				
Asian	16	16	15	13
European	77	77	74	77
Māori	3	3	6	5
Pacific Islander	1	1	3	3
Other	3	3	2	3
**Year of study**				
First	78	78	84	82
Second	13	13	11	11
Third	4	4	3	4
4th year or above	5	5	3	3

**Table 2 ijerph-17-00836-t002:** Alcohol-related harms in the preceding 4 weeks attributed by respondents to their own drinking.

	2004 n = 1360	2014 n = 1622		Change ^2^	
	*%*	*%*	*Adj%* ^1^	*%*	*95% CI*
Suffered a hangover	59	55	59	0.8	(−4.3, 6.0)
Had a heated argument	13	8.5	10	−3.1	(−6.0, −0.1)
Had a blackout	40	38	43	2.8	(−0.2, 5.7)
Had sex they later regretted	12	9.7	11	−1.1	(−2.8, 0.5)
Had sex they were unhappy about at the time	6.0	3.7	4.0	−2.1	(−3.9, −0.3)
Had an emotional outburst	23	22	23	0.7	(−2.7, 4.1)
Was unable to pay bills	5.7	2.2	2.6	−3.0	(−4.2, −1.9)
Had unsafe sex	5.5	5.6	6.3	0.7	(−1.4, 2.9)
Vomited	32	29	33	0.6	(−3.2, 4.3)
Committed an act of vandalism	7.1	2.0	2.7	−5.2	(−7.7, −2.7)
Stole private or public property	11	3.8	4.5	−6.5	(−8.2, −4.7)
Was physically aggressive towards someone	10	4.6	5.3	−4.8	(−6.9, −2.6)
Was removed from a pub/club	11	5.0	6.0	−4.5	(−6.9, −2.1)

^1^. Adjusted for differences in gender, age, and ethnicity composition, and past-year hazardous drinking, between 2004 and 2014 respondents; ^2^. Change between the 2004 raw estimate and the 2014 adjusted estimate.

**Table 3 ijerph-17-00836-t003:** Alcohol-related harms in the preceding 4 weeks attributed by respondents to others’ drinking.

	2004 n = 1629	2014 n = 1869		Change ^2^	
	*%*	*%*	*Adj%* ^1^	*%*	*95% CI*
Experienced an unwanted sexual advance	14	8.7	8.9	−4.6	(−7.1, −2.1)
Found vomit in the halls or bathrooms	43	39	40	−3.4	(−14, 7.1)
Had a serious argument	9.3	7.7	8.6	−0.7	(−2.4, 0.1)
Had their property damaged	11	9.3	9.7	−1.1	(−3.3, 1.2)
Had their study or sleep interrupted	67	59	59	−7.9	(−15, −1.2)
Had to ‘baby−sit’ a drunk student	44	55	55	11.3	(6.7, 15.8)
Was a victim of sexual assault or date rape	1.0	0.4	0.4	−0.6	(−1.2, −0.1)
Was insulted or humiliated	16	15	16	0.5	(−2.1, 3.0)
Was pushed, hit or otherwise assaulted	6.9	5.6	6.0	−0.9	(−2.6, 0.7)

^1^. Adjusted for differences in gender, age, and ethnicity composition, and past-year hazardous drinking, between 2004 and 2014 respondents; ^2^. Difference between the 2004 raw estimate and 2014 adjusted estimate.

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
