# Peer review of "Alcohol Harms over a Period of Alcohol Policy Reform: Surveys of New Zealand College Residents in 2004 and 2014"

_ijerph, 2020, doi:10.3390/ijerph17030836_

Round 1
Reviewer 1 Report
Dear authors,
I have reviewed the resubmitted manuscript in which you have carefully addressed all comments and suggestions. It is a well structured and written manuscript that can be accepted without further revisions.
Reviewer 2 Report
The authors have satisfactorily addressed my concerns.
Reviewer 3 Report
no comments
This manuscript is a resubmission of an earlier submission. The following is a list of the peer review reports and author responses from that submission.
Round 1
Reviewer 1 Report
Dear authors,
This is an interesting manuscript addressing the problem of alcohol consumption in college. It will be greatly improved through some minor interventions:
1.Introduction can be improved
present the prevalence of alcohol drinking and prevalence of drinking to intoxication among college students and possibly contrast with prevalence in general population. define why it is a major concern refer previous studies if existing and point out possible gaps which you have addressed in the current study highlight the strengths and novelty of your study2.Discussion, 3rd paragraph, 1st line:
The timing of the surveys in the same weeks of the academic year ine ach of the surveys,
Should read:
the timing of the surveys in the same weeks of the academic year in each of the surveys,
3.Conclusion
Since "baby-sitting" of drunk students increased in the frame of decreased reporting alcohol related problems(2004-2014), it could be included in the conclusion in addition to the strategies of decreased availability and promotion of alcohol in campus.
Author Response
(1) Introduction can be improved
present the prevalence of alcohol drinking and prevalence of drinking to intoxication among college students and possibly contrast with prevalence in general population. define why it is a major concern refer previous studies if existing and point out possible gaps which you have addressed in the current study highlight the strengths and novelty of your study
RESPONSE: We have added the following as the second paragraph of the introduction:
“In studies making the comparison, which were mainly been performed in the USA in the early 2000s, college students in their late teens and early 20s were found to consume a greater volume of alcohol and had a higher prevalence of consuming ≥5 drinks on a single occasion (“binge drinking”) than thein non-student peers [2]. College students were also more likely to report symptoms of acute problems from their drinking (“alcohol abuse”) [3]. A New Zealand study estimated that compared with their non-student peers, university students had twice the prevalence of scoring ≥8 (“hazardous drinking”) and three times the prevalence of scoring ≥15 (“harmful drinking”), on the Alcohol Use Disorders Identification Test [4].”
And the following in the final paragraph:
“While drinking behaviour is an important outcome by which to judge the effects of changes in policy, it is arguably a means to the end of reducing the incidence and severity of various harms. Vandalism, physical aggression, and sexual assault are what motivated efforts to make the campus and surrounding environment safer for students and other members of the community. Accordingly, in this study we estimated change in the prevalence of a range of health, social, and legal consequences of risky drinking, in the same population, between 2004 and 2014.”
We have added further material in response to later reviewer comments which we detail below.
(2) Discussion, 3rd paragraph, 1st line:
The timing of the surveys in the same weeks of the academic year ine ach of the surveys,
Should read: the timing of the surveys in the same weeks of the academic year in each of the surveys,
RESPONSE: Done.
(3) Conclusion
Since "baby-sitting" of drunk students increased in the frame of decreased reporting alcohol related problems(2004-2014), it could be included in the conclusion in addition to the strategies of decreased availability and promotion of alcohol in campus.
RESPONSE: We have added the following in the paragraph immediately before the conclusion (p.10):
“Notably, the increase in “baby-sitting” of drunk students despite decreasing drinking to intoxication over the same period, suggests success in the colleges’ efforts to encourage students to look after each other when drinking.”
Reviewer 2 Report
Kypri et al. has written a short report on the survey results of college residents regarding drinking with a ten-year time interval. The report is interesting and offers insight to the issue of alcoholism and effect of policy reform in a regional population. I recommend acceptance as it is.
I would like to make it a minor revision then, and add the following comments: The current manuscript, especially the introduction, is relatively short. It would be good to further elaborate on the details of the alcohol policy reform in New Zealand, and to introduce more about other studies that investigated the effects of such reform on the behavior of population groups in the country. The readers can then better comprehend and appreciate the connection between the results of the current survey and the policy reform.
Author Response
(4)
Kypri et al. has written a short report on the survey results of college residents regarding drinking with a ten-year time interval. The report is interesting and offers insight to the issue of alcoholism and effect of policy reform in a regional population. I recommend acceptance as it is.
I would like to make it a minor revision then, and add the following comments: The current manuscript, especially the introduction, is relatively short. It would be good to further elaborate on the details of the alcohol policy reform in New Zealand, and to introduce more about other studies that investigated the effects of such reform on the behavior of population groups in the country. The readers can then better comprehend and appreciate the connection between the results of the current survey and the policy reform.
RESPONSE: The reforms we refer to are not changes at the national or provincial level, but rather, in university policy. We have added the following on pp.3-4:
“We attributed those changes to a combination of deliberate efforts by the University to reduce the promotion of alcohol and student misconduct [6], and market conditions that caused the closure of student pubs in the area surrounding campus [5]. These efforts and conditions were independent of more recent alcohol policy reforms in New Zealand which are unlikely to have had a significant impact on student drinking in 2013-14 due to the phased implementation of the reforms (some changes came into effect after the 2014 survey) and the absence of a local alcohol policy in Dunedin until 2019 [7, 8].”
Reviewer 3 Report
First of all I want to congratulate the authors for the work done, the analysis of an intervention not only in the consequences, but also in the consequences of consumption. The excessive consumption of alcohol in university students and its consequences is a public health problem present in different countries and it is therefore, of interest knowing different experiences of interventions to adress it
I wanted to make some indications to the authors that I believe would improve the manuscript.
Introduction: missing information on other experiences of interventions in the reduction of alcohol consumption by university students. They do not present a comprehensive review of the literature in this regard.
methology:
I think it would be interesting to give explicit information about the measured variables and the methodology in general so that the reader dont have to resort to the references of previous studies.
Results:
I consider that a descriptive table of the population that answers the survey as the first table of the manuscript would be interesting and would improve it.
Discussion:
I like how they have assessed the different results obtained. However, as in the introduction, I miss a review of the scientific evidence regarding interventions in the university population to compare the results of the present study.
It is explain at the discussion that there was a decrease of on-licenced premises at the area, could not be possible to obtain a variable that measured this and take it into account in the analysis? Although it is somewhat simple, for example the density of places where alcohol is consumed in 2004 and 2014 in these areas.
Author Response
(5)
Introduction: missing information on other experiences of interventions in the reduction of alcohol consumption by university students. They do not present a comprehensive review of the literature in this regard.
RESPONSE: We have added the following on p.3:
“The New Zealand experience contrasts with findings from surveys of nationally representative samples of college students between 1993 and 2001 in the USA. Wechsler and colleagues [9] found no change in the prevalence of heavy episodic drinking among students despite participants reporting increased exposure to university interventions over this 8-year period. These interventions primarily were education-based programs on the health effects of alcohol, provision of information on seeking help for alcohol problems, dissemination of institutional alcohol rules and the penalties for breaching these, and a slight increase in the use of punitive measures (i.e., fines, community service, compulsory education program attendance and treatment referrals). Previous interventions found to be effective among university students have involved authorities external to the university increasing their enforcement of state liquor laws, however efforts are not applied widely or consistently.
[10-12].”
(6)
methology:
I think it would be interesting to give explicit information about the measured variables and the methodology in general so that the reader dont have to resort to the references of previous studies.
RESPONSE: We provide this information in the Analysis section on p.5.
(7)
Results: I consider that a descriptive table of the population that answers the survey as the first table of the manuscript would be interesting and would improve it.
RESPONSE: Done. Please see Table 1, on p.13.
(8)
Discussion:
I like how they have assessed the different results obtained. However, as in the introduction, I miss a review of the scientific evidence regarding interventions in the university population to compare the results of the present study.
RESPONSE: We have added the following in the Discussion, on p.11:
“The findings are broadly concordant with previous research has shown that policies restricting the availability and promotion of alcohol are the most effective at reducing alcohol consumption and harm among the general population [25]. There is evidence that these policies are also effective among university students, particularly when implemented alongside other strategies, although it is not clear what the ideal package of strategies might be [26].”
(9)
It is explain at the discussion that there was a decrease of on-licenced premises at the area, could not be possible to obtain a variable that measured this and take it into account in the analysis? Although it is somewhat simple, for example the density of places where alcohol is consumed in 2004 and 2014 in these areas.
RESPONSE: We have added the following in the Discussion, on p.8:
“Quantifying the harm associated with the closures would be problematic given that the change in exposure would be relatively small. However, the pubs that closed were “student pubs” that had been frequented by many generations of students. Their closure may have had disproportionate effects that could not be captured by a simple test of correlation.”
Reviewer 4 Report
The paper reviews the changes in the prevalence of harms attributed by students after alcohol consumption. The current manuscript concerns a very important and contemporary problem, because alcohol is widely abused. Moreover, drinking of alcohol at young age is related not only with anti-social behavior but also increases risk of addiction and neurological impairments in adulthood.
The topic presented in the manuscript is interesting and present-day. The manuscript is well written in accordance with the requirements of the journal, therefore I have only a few minor comments:
The Authors should unify the font throughout the manuscript. In my opinion too many study participants were women. The participants of both sexes should be in a comparable number while in this study two thirds are women. This is extremely important since we observe different drinking patterns, addiction and alcohol-related behavior in both sexes. How the Authors explain this fact? The Authors should explain exactly where the Adj% value in tables 1 and 2 came from? How it was calculated? The wider significance of the data being processed is also controversial. Especially that research was conducted in 2004 and 2014 and will be published only in 2019. The Authors should also indicate exactly what benefits result from conducted policy reforms and what can be changed in the future to improve safety.Author Response
(10)
The Authors should unify the font throughout the manuscript.
RESPONSE: Done.
(11)
In my opinion too many study participants were women. The participants of both sexes should be in a comparable number while in this study two thirds are women. This is extremely important since we observe different drinking patterns, addiction and alcohol-related behavior in both sexes. How the Authors explain this fact?
RESPONSE: We have added the following on p.8:
“It is noteworthy that two thirds of the respondents in the 2014 survey were women. This reflects the fact that there are more women at the University and residing in the halls, and in that women were more likely to respond. The study aimed to recruit a sample to represent the student population. Our previous analysis of selective non-response in this population suggests that it is likely to have biased estimates of the prevalence of alcohol-related harm to a small extent [17]. Relatedly, drinking patterns in this population are similar by gender [18].”
(12)
The Authors should explain exactly where the Adj% value in tables 1 and 2 came from? How it was calculated?
RESPONSE: We state on p.5 that adjusted prevalence estimates were obtained using the margins command in Stata. We have added a reference to this command.
(13)
The wider significance of the data being processed is also controversial. Especially that research was conducted in 2004 and 2014 and will be published only in 2019. The Authors should also indicate exactly what benefits result from conducted policy reforms and what can be changed in the future to improve safety.
RESPONSE: As a mainly descriptive study, the findings do not support strong causal inferences. Accordingly, we consider our conclusion to be appropriately cautious.